# Enzymatically Synthesized Ginsenoside Exhibits Antiproliferative Activity in Various Cancer Cell Lines

**Sumangala Darsandhari [1], Biplav Shrestha [1], Ramesh Prasad Pandey [1,2] , Sanghun Lee [1], Hye Jin Jung [1] , Yeon Ju Kim [3] and Jae Kyung Sohng [1,2,\***

[1]   Department of Life Science and Biochemical Engineering, SunMoon University, 70 Sunmoon-ro 221, Tangjeong-myeon, Asan-si, Chungnam 31460, Korea; sumangala@sunmoon.ac.kr (S.D.); biplav@sunmoon.ac.kr (B.S.); pandey@sunmoon.ac.kr (R.P.P.); shlee@sunmoon.ac.kr (S.L.); poka96@sunmoon.ac.kr (H.J.J.)

[2]   Department of Pharmaceutical Bioengineering, SunMoon University, 70 Sunmoon-ro 221, Tangjeong-myeon, Asan-si, Chungnam 31460, Korea

[3]   Department of Oriental Medicinal Biotechnology, College of Life Science, Kyung Hee University, 1732, Deogyeong-daero, Giheung-gu, Yongin-si, Gyeonggi-do 17104, Korea; yeonjukim@khu.ac.kr

\*   Correspondence: sohng@sunmoon.ac.kr; Tel.: +82-(41)-530-2246; Fax: +82-(41)-530-8229

**Abstract:** A glycoside derivative of compound K (CK) was synthesized by using a glycosyltransferase, and its biological activity was tested against various cancer-cell lines. A regiospecific, β-1,4-galactosyltransferase (LgtB) converted 100% of 0.5 mmol CK into a galactosylated product in 3 h. The structure of the synthesized derivative was revealed with high performance liquid chromatography, mass spectroscopy, as well as nuclear magnetic resonance analyses, and it was recognized as 20-O-β-D-lactopyranosyl-20(S)-protopanaxadiol (CKGal). Out of the four cancer-cell lines tested (gastric carcinoma (AGS), skin melanoma (B16F10), cervical carcinoma (HeLa), and brain carcinoma (U87MG)), CKGal showed the best cytotoxic ability against B16F10 and AGS when compared to other ginsenosides like compound K (20-O-β-D-glucopyranosyl-20(S)-protopanaxadiol), Rh2 (3-O-β-D-glucopyranosyl-20(S)-protopanaxadiol), and F12 (3-O-β-D-glucopyranosyl-12-O-β-D-glucopyranosyl-20(S)-protopanaxadiol). Thus, the synthesized derivative (CKGal) is a pharmacologically active ginsenoside.

**Keywords:** anticancer; β-1,4-galactosyltransferase; Compound K; ginsenoside

## 1. Introduction

Ginseng is one of the most commonly consumed traditional medicines in many Asian countries. It has been used for many years to maintain physical strength and increase resistance to stress or aging [1]. One of the major pharmacologically active components of ginseng, ginsenosides, have numerous biological activities, such as anti-inflammatory, anticancer, antiaging, antiviral, and antitumor activities [2–6]. Ginsenosides are triterpenoid glycosides containing dammarane and are generally divided into three ginsenoside groups: protopanaxadiol (PD), protopanaxatriol (PT), and oleanolic acid. These compounds have been reported to exhibit a wide variety of physiological and pharmacological properties. $F_2$, $Rh_2$, $Rg_3$, and compound K (CK, 20-O-β-D-glucopyranosyl-20(S)-protopanaxadiol) are rare ginsenosides, which are present either at low concentrations or absent in ginseng [7,8]. The variations in the linkage positions and type of sugar result in different pharmacological activities in these ginsenosides. Research has shown that the rare ginsenosides show relatively strong anticancer activities [9]. Compound K (CK) is a key metabolite

of ginseng saponin, which is a PD type ginseng, synthesized after their oral administration by intestinal bacteria [10]. It is believed that CK is absorbed from the intestine [11] and impedes the proliferation of many tumor cells, such as liver cancer cells [12], lung carcinoma [13], colorectal carcinoma [14], and glioblastoma [15] by preventing cell growth and inducing apoptosis. CK was also reported to improve the effectiveness of anticancer drugs in some drug-resistant cancer cells [16].

There has been a continuous search for plant products and the derivatives of such products that may have health benefits for humans. Ginsenoside transformation has been conducted by physical, chemical, and biological methods [1,17–20]. An active ginseng metabolite with different or better bioactivity may be produced by modifying its structure in various ways [21]. Therefore, several studies of the biotransformation of major ginsenosides have been done on microorganisms for decades [21–23]. However, transformation by chemical methods is prone to many side reactions, such as hydroxylation, hydration, and epimerization [24]. Regiospecific glycosylation is an effective way to achieve structurally diversified compounds using a simple reaction process, such as an enzymatic method rather than a complicated method using chemicals. Glycosyltransferase enzymes are mainly utilized for the synthesis of complex oligosaccharides and glycoconjugates [25,26]. Glycosylation may improve the biological properties, enhance the physical and chemical stability and solubility, and amend the subcellular localization of the parent compound. In the present study, we have synthesized a derivative of CK enzymatically by using a *β*-1,4-galactosyltransferase (LgtB) and have evaluated the anti-proliferative activity of the newly synthesized compound on gastric cancer-cell lines (AGS), skin melanoma (B16F10), cervical carcinoma (HeLa), and brain carcinoma (U87MG). The activity of the synthesized compound was investigated and compared with other ginsenosides like $Rh_2$, CK, and 3-*O*-*β*-D-glucopyranosyl-12-*O*-*β*-D-glucopyranosyl-20(S)-protopanaxadiol (F12 or F2) (Figure 1). Previously, a similar report on the modification of CK to galactose conjugated derivative was performed using bovine GalT [26], but its anticancer activity has not yet been tested.

**Figure 1.** Chemical structures of the protopanaxadiol and ginsenoside derivatives of protopanaxadiol.

## 2. Results and Discussion

In this experiment, LgtB was expressed as an N-terminal $His_6$-tagged protein heterogeneously in *E. coli* BW25113 (DE3). The expression of the soluble protein was best when 0.5 mmol isopropyl-*β*-D-thiogalactopyranoside (IPTG) was used for the induction of bacterial culture.

The ginsenoside CK was modified by using purified LgtB protein and attaching galactopyranosyl moiety at the 4-OH position of glucose in CK. By using a *β*-1,4 galactosyltransferase, the glycosylation of CK was quantitatively analyzed by an ultrahigh-performance liquid-chromatography photo-diode array (UHPLC-PDA) analysis. The reaction mixture yielded a peak (P) which appeared before a retention time ($t_R$) of a substrate peak (S). The control reaction performed with a soluble lysate from *E. coli* Bw25113 (DE3) expressing LgtB in pET24 vector, which was not induced, did not show any new peak. The peaks of the substrate and product were observed at $t_R$ of ~ 38.1 and 36.2 min, respectively (Figure 2a). A total of 0.5 mmol CK was transformed completely into the product within 3 h. The UV spectra of the substrate and that of the product, as shown in Figure 2b, were alike.

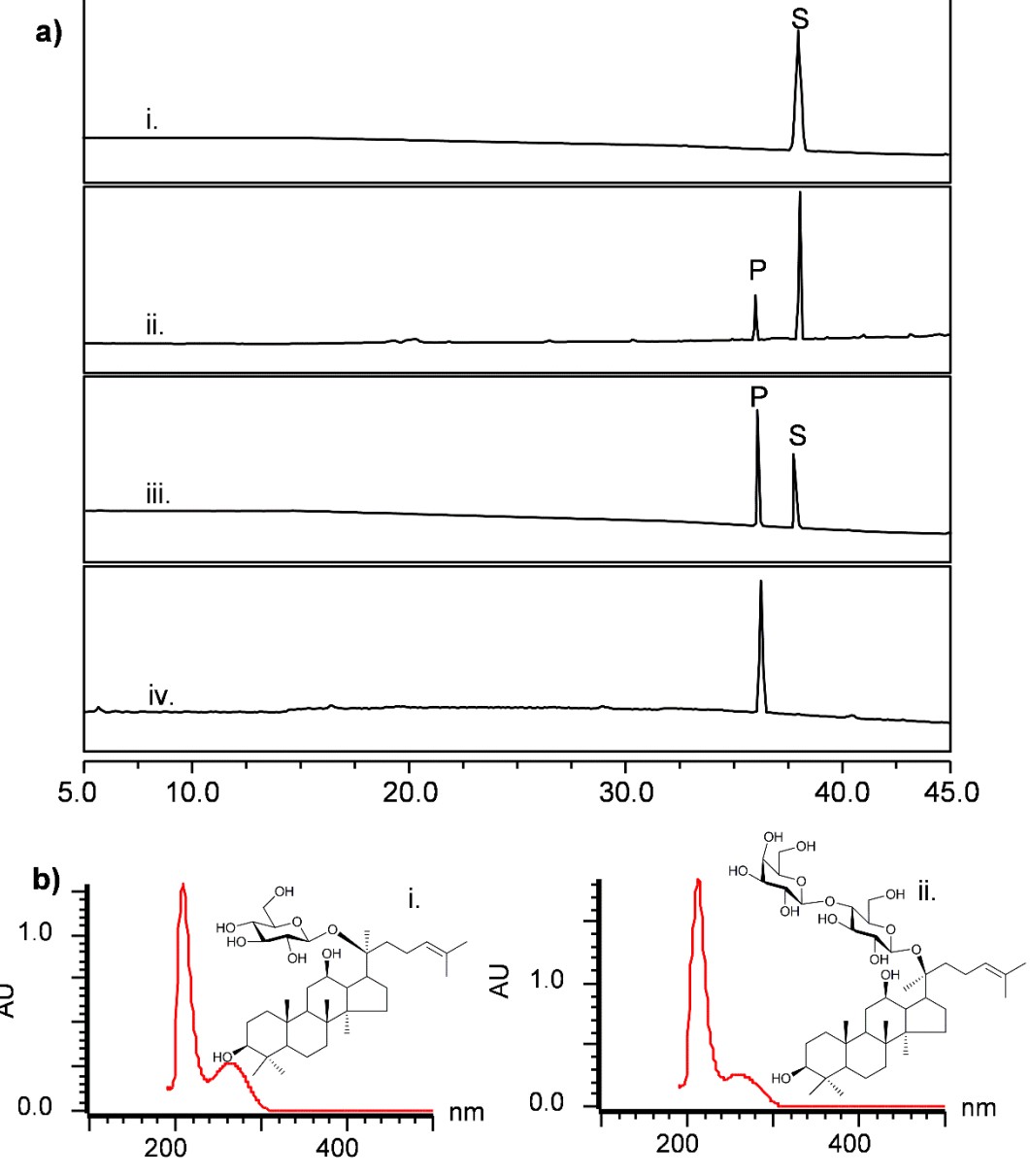

**Figure 2.** HPLC of the glycosylation reaction of compound K (CK) by purified recombinant LgtB. (**a**) HPLC analysis of the reaction of CK; Control (i), Reaction at 0.5 h (ii), reaction at 1 h (iii), and reaction at 3 h (iv). (**b**) UV maxima of (i) CK (S), and (ii) CKGal (P). S refers to substrate and P refers to product.

We further elucidated the chemical structure of the metabolite by its accurate mass and mass-spectral fragmentation pattern and by 1D and 2D NMR analyses. The exact calculated mass

of [CK + galactose + H]+ was 785.5051 Da, and the product peak displayed a similar mass fragment *m/z*+ of 785.5067 (Figure 3). To further confirm the structure of the product peak, it was purified by preparative-HPLC and prepared for NMR analysis as described in the materials and methods section. The $^1$H-NMR spectrum of the reaction product showed two proton signals representing beta ($\beta$) configurations for H-1 of the glucopyranosyl and galactopyranosyl moieties, which appeared at $\delta$ 5.12 ppm (1H, *d*, *J* = 7.8 Hz) and $\delta$ 5.04 ppm (1H, *d*, *J* = 7.8 Hz), respectively. Other proton signals of sugar moieties were seen in the region from $\delta$ (3.0–4.0) ppm (Supplementary Figure S1A). When the $^{13}$C-NMR spectrum of the reaction product was compared with that of CK, six new signals for C-1″ to C-6″ of galactopyranoside were present between $\delta$ 60 ppm and $\delta$ 105.57 ppm. Other signals of product exactly matched with that of CK (Supplementary Figure S2). The signal for the anomeric carbons of glucopyranosyl and galactopyranosyl moieties appeared at $\delta$ 97.65 ppm and $\delta$ 105.57 ppm (Supplementary Figure S1B), respectively. Their correlations were measured in heteronuclear single-quantum correlation (HSQC) for approving the position at which sugar is conjugated (Supplementary Figure S1C). A downfield glycosylation shift of $^{13}$C (~11 ppm) of the C4′ carbon of glucose was observed, which indicated the attachment of a galactosyl moiety to the C4′-OH of CK (Table 1). As seen in Supplementary Figure S1D, in the heteronuclear multiple-bond correlation (HMBC), a long-range correlation between C4′ of the CK backbone (C4 of glucosyl moiety of CK, $\delta$ 81.96) and anomeric proton of the galactosyl moiety (H1″, $\delta$ 5.04) further confirmed that the product was 20-*O*-$\beta$-D-lactoside of PD. The correlation between 1′-H at $\delta$ 5.12 ppm and 4′-C at $\delta$ 97.65 (C-4 of glucopyranosyl moiety), and 1″-H at $\delta$ 5.04 ppm and 4″-C at $\delta$ 105.58 (C-4 of galactopyranosyl moiety) was found in the HMBC (Supplementary Figure S1D). Cumulatively, the evidence from several analytical data, allowed us to conclude that the glycosylated product obtained from the galactosylation reaction of CK was 20-*O*-[$\beta$-D-lactopyranosyl]-20(S)-protopanaxadiol.

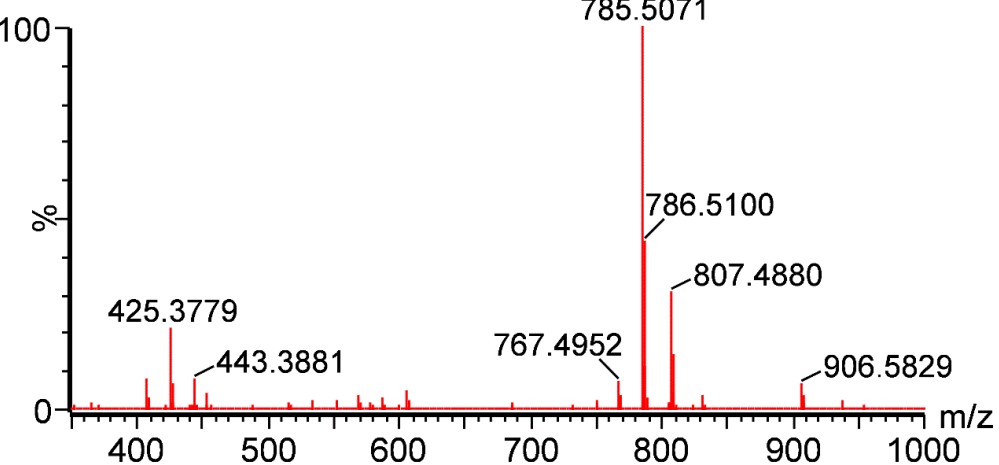

**Figure 3.** High-resolution quadrupole-time-of-flight electrospray-ionization mass spectrometry (HR-QTOF ESI/MS) analysis showing the product of CK. Mass analysis of product ([M + H]+ *m/z*+ 785.5060) confirmed that the product was a monogalactoside derivative of CK ($C_{42}H_{72}O_{13}$, calculated molecular weight, [M + H]+ *m/z*+ 785.505).

There has been a continuous search for active and novel anticancer agents over the past decade. Ginsenoside, the chief pharmacologically active component of ginseng, produces a collection of pharmacological responses. The compounds $Rh_2$, F12, CK, and CKGal were tested for in vitro cytotoxicity by 3-(4,5-dimethylthiazol-2-yl)-2,5-diphenyltetrazolium bromide (MTT) colorimetric assay against four different cancer-cell lines. A recent study has shown that F12 effectively repressed the spread of various cancer-cell lines, showing much better in vitro cytotoxicity than Rg3 [27]. The ginsenoside CK has gained a particular interest in cancer chemoprevention and treatment. Lee et al. (2000) showed that CK induced an anticancer or anti-metastatic activity by preventing tumor invasion

or averting chromosomal activation and tumorigenesis [17]. CK in combination with γ-ray radiation was suggested for cancer therapy by Chae et al. (2009) [12]. Likewise, Zhang et al. (2013) revealed for the first time the participation of endoplasmic reticulum stress in the induction of apoptosis by CK in HT-29 cells [2]. Our results indicated that CKGal showed good cytotoxic activities compared to three other compounds tested. The cell viability of gastric carcinoma (AGS), skin melanoma (B16F10), cervical carcinoma (HeLa), and brain carcinoma (U87MG) was reduced to approximately 8.7%, 2.6%, 7.3%, and 13.7%, respectively, compared to controls, when treated with 200 μmol of CKGal (Figure 4). The IC$_{50}$ values (50% inhibitory concentration) of CKGal for AGS, B16F10, HeLa, and U87MG cells were 4.487, 22.4, 40.38, and 40.38 μmol, respectively. CK inhibited AGS and B16F10 cell lines with an IC$_{50}$ value of 25.99 μmol and 35.17 μmol, respectively. Rh$_2$ showed better activity against all the cell lines tested than F12, and showed the best anticancer activity in U87MG glioblastoma cells, whereas CK was most effective against HeLa cell lines (Supplementary Table S1). The ginsenoside CK is a monoglycoside derivative of protopanaxadiol, whereas F12 and CKGal are diglycoside derivatives of protopanaxadiol. Our study with ginsenosides has shown that the in vitro cytotoxicity of CKGal against different cancer-cell lines was much higher than that of F12, which was reported to have a much better cytotoxicity than Rg3, another diglycoside derivative of protopanaxadiol. [28]. However, the fact that ginsenoside CKGal exhibited stronger inhibitory activity against cancer cells than ginsenoside CK is contrary to a previous finding, which says that the presence of more sugar moieties in ginsenosides decreases its inhibitory activity against cancer cells [28]. The improvement in such a biological property may have resulted from the presence of a different sugar moiety (lactose) instead of glucose in the protopanaxadiol. While ginsenosides exhibit therapeutic activity, attachment of the sugar units enhances the solubility of compounds. Glycosides act as a key component for improving various pharmacological actions, such as anticancer, antifungal, and antitumor activities [29].

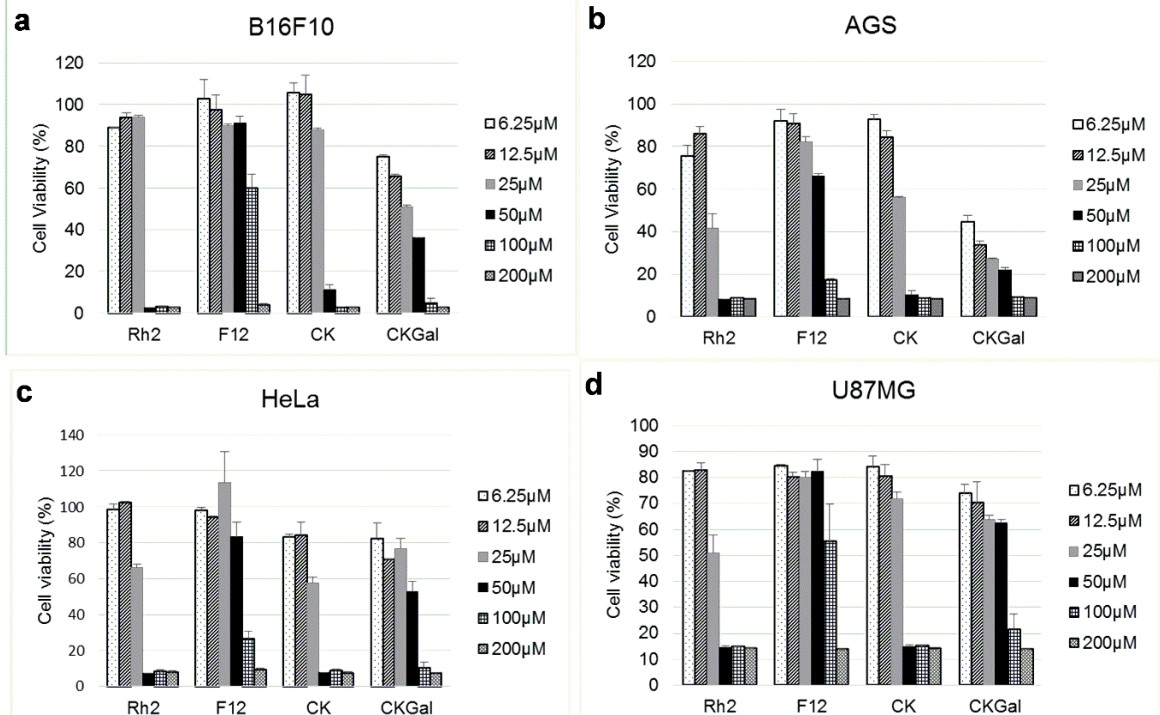

**Figure 4.** The growth inhibitory effect of ginsenosides Rh2, F12, CK, and CKGal on cancer cells. (**a**) B16F10 (skin cancer), (**b**) AGS (gastric cancer), (**c**) HeLa (cervical cancer), and (**d**) U87MG (brain cancer). The cells were treated with increasing concentrations of compounds (6.25, 12.5, 25, 50, 100 and 200 μmol) for 72 h and cell growth was measured by MTT assay.

The potential molecular mechanism of CKGal has yet to be explored. However, components of ginseng have been found to induce cell cycle arrest in the G1 phase by upregulating the expression of p21 [6]. Extracts from ginseng have been found to inhibit cancer cell growth, both through suppression of the MAPKs pathway [30] and through activation of MAPKs [31]. Ethanol extract of ginseng was found to downregulate the expression of proliferating cell nuclear antigens (PCNAs) in vivo, and the activated form of Stat-3 in vitro [31]. Compound K is known to upregulate the expression of p21 and activate *c*-jun N-terminal kinase (JNK) for cell cycle arrest in the G$_1$ phase in human monocytic leukemia cells [32]. Although our research lacks study in the mechanism of action of such molecules, the potential mechanism of action of such compounds may be involved with the activities associated in the regulation of cell cycle, apoptosis, or proliferation of cells. Previous studies have shown an enhanced anticancer activity of ginsenosides when the sugar units are found in the C-3 or C-20 position in comparison to sugar units at the C-6 position [6]. In this study, CK and Rh2, which contain sugar in the C-20 and C-3 position, respectively, showed quite similar activity against all the cancer cells tested. The anticancer activity of both of these ginsenosides was better than that of F12. CKGal showed better anticancer activity in the case of B16F10 and AGS cancer cell lines when compared with CK. However, the activity was not improved in HeLa and U87MG cancer cell lines. It is now an established fact that the number of sugar moieties and different substituent groups in the side chain of the dammarane saponin or aglycone protopanaxadiol affects the activity of the ginsenosides against different cancer cell lines. On the other hand, the addition of sugar moieties increases the water solubility of the compound. Hence, it may be possible that it is easier for the compound to reach the target because of the increase in its solubility. We anticipate that this preliminary study on CKGal biosynthesis and screening of cancer cell line growth inhibitory activity will be helpful for evaluating the structure-activity relationship studies against cancers with other ginsenosides in the near future.

**Table 1.** $^{13}$C-NMR of ginsenosides, 20-*O*-[*β*-D-glucopyranosyl]-20(*S*)-protopanaxadiol (CK), and 20-*O*-[*β*-D-lactopyranosyl]-20(*S*)-protopanaxadiol (CKGal) in Pyridine-*d*$_5$.

| Carbon Site | CK (ppm) | CKGal (ppm) |
|---|---|---|
| 1 | 39.42 | 39.32 |
| 2 | 28.29 | 27.97 |
| 3 | 78.05 | 77.83 |
| 4 | 39.58 | 39.13 |
| 5 | 56.37 | 56.11 |
| 6 | 18.79 | 18.52 |
| 7 | 35.18 | 34.92 |
| 8 | 40.08 | 39.82 |
| 9 | 50.32 | 50.05 |
| 10 | 37.37 | 37.1 |
| 11 | 30.79 | 30.61 |
| 12 | 71.7 | 70.8 |
| 13 | 49.55 | 49.43 |
| 14 | 51.43 | 51.18 |
| 15 | 30.99 | 31.13 |
| 16 | 26.67 | 26.38 |
| 17 | 51.58 | 51.31 |
| 18 | 16.05 | 15.78 |
| 19 | 16.38 | 16.12 |
| 20 | 83.29 | 83.23 |

**Table 1.** *Cont.*

| Carbon Site | CK (ppm) | CKGal (ppm) |
| --- | --- | --- |
| 21 | 22.33 | 22.05 |
| 22 | 36.21 | 35.77 |
| 23 | 23.19 | 22.77 |
| 24 | 125.98 | 125.64 |
| 25 | 130.90 | 130.78 |
| 26 | 25.78 | 25.61 |
| 27 | 17.77 | 17.55 |
| 28 | 28.70 | 28.46 |
| 29 | 16.36 | 16.80 |
| 30 | 17.42 | 17.48 |
| 1′ | 98.27 | 97.65 |
| 2′ | 75.13 | 76.01 |
| 3′ | 79.38 | 77.83 |
| 4′ | 70.15 | 81.96 |
| 5′ | 78.31 | 77.08 |
| 6′ | 62.93 | 62.04 |
| 1″ | | 105.57 |
| 2″ | | 74.95 |
| 3″ | | 72.28 |
| 4″ | | 69.82 |
| 5″ | | 77.00 |
| 6″ | | 61.84 |

## 3. Materials and Methods

### 3.1. Chemicals and Strains

Compound K was obtained from Ilhwa Co., Ltd (Guri, Korea). Rh2 (3-*O*-*β*-D-glucopyranosyl-20(S)-protopanaxadiol) and F12 (3-*O*-*β*-D-glucopyranosyl-12-*O*-*β*-D-glucopyranosyl-20(S)-protopanaxadiol) were obtained from Professor Yeon Ju Kim at Kyung Hee University. HPLC-grade acetonitrile and water were bought from Mallinckrodt Baker (Phillipsburg, NJ, USA). Standard UDP-*α*-D-galactose and IPTG were purchased from GeneChem (Daejeon, South Korea). Roswell Park Memorial Institute 1640 medium (RPMI) was purchased from Invitrogen, USA. All other chemicals used were of high analytical grade and commercially available.

### 3.2. Expression, Purification, and Quantification

The gene *β*-1,4-galactosyltransferase (LgtB, GenBank: AAM33872.1) synthesized from GenScript (USA) had *Nde*I and *Hin*dIII as restriction sites and was cloned into a vector pET24ma. For the preparation of recombinant protein, pET24ma-LgtB [33] was transformed into *E. coli* Bw25113 (DE3). The recombinant *E. coli* was cultured in Luria–Bertani (LB) medium which contained 50 mg $l^{-1}$ kanamycin at 37 °C until the absorbance ($OD_{600}$) reached between 0.6 and 0.8. Then the cells were induced by adding IPTG to a final concentration of 0.5 mmol via incubation at 20 °C for approximately 20 h. The cells were collected by centrifugation at 850× *g* at 4 °C and washed with 50 mmol Tris/HCl buffer (pH 7.5). After that, the cells were sonicated, and the clear lysate was collected by centrifuging at 13,000× *g* for 30 min at 4 °C. The protein was purified using TALON metal nickel affinity resin (Takara Bio, Shiga, Japan). The crude lysate of the soluble protein was mixed into the resin equilibrated with a buffer which contained 100 mmol Tris/HCl (pH 7.5) and 300 mmolNaCl. This mixture was

slightly agitated in ice for 30 min before elution with imidazole. The fractions collected were examined in 12% sodium dodecyl sulfate-polyacrylamide gel electrophoresis (SDS-PAGE), and the enzyme concentration was measured by Bradford's method. The protein thus obtained was kept at -70 °C until used for in vitro reactions.

### 3.3. Enzymatic Reaction Conditions

A 25 mmol stock solution of CK, made in dimethyl sulfoxide (DMSO), was used for in vitro reactions. At first, 0.3 mmol of CK was used as a substrate in a 200 µl reaction. The reaction was performed in 50 mmolphosphate buffer (pH 7.5), with 10 mmol $MnCl_2$, 1 mmol UDP-$\alpha$-D-galactose, and 20 µg/mL of purified enzyme. The reaction was incubated at 37 °C for 1 h and was terminated by adding a double volume of butanol. It was vortexed, centrifuged at 13,000× *g* for 5 min, filtered through a 0.2 µm filter, and subjected to UHPLC-PDA and high-resolution quadrupole-time-of-flight electrospray-ionization mass spectrometry (HR-QTOF ESI/MS) analyses.

The reaction was scaled up to 50 mL total volume, containing 30 µg/mL purified LgtB, 0.5 mmol CK, 2 mmol UDP-$\alpha$-D-galactose, 50 mmol phosphate buffer (pH 7.5), and 10 mmol $MnCl_2$ at 37 °C for 3 h. It was terminated as described above.

### 3.4. Analytical Procedures

A $C_{18}$ column (Mightysil RP-18 GP) (4.6 mm × 250 mm, 5 µm) was used for the reverse-phase UHPLC-PDA analysis. It was performed using HPLC-grade water containing 0.1% formic acid (A) and acetonitrile (B) at a flow rate of 1 mL/min at 210 nm. Chromatographic separation was carried out at 35 °C. The program used was: 0–10 min, 20% B; 10–20 min, 20%–45% B; 20–40 min, 45%–80% B; 40–55 min, 80%–100% B; and 55–60 min, 100%–20% B.

The HR-QTOF ESI/MS analysis was performed using an ACQUITY (UPLC, Waters Corp., Billerica, MA, USA) column coupled with a SYNAPT G2-S (Water Corp.) in positive-ion mode. The compound was purified using preparative-HPLC equipped with a $C_{18}$ column (YMC-Pack ODS-AQ) (250 × 20 mm I.D., 10 µm) connected to a UV detector (SPD-20A) with a 110-min binary program: 0–25 min, isocratic 15% B; 25–60 min, 15%–45% B; 60–80 min, 45%–80% B; 80–100 min, 80%–100% B; and 100–110 min, 100%–100% B at a flow rate of 10 mL/min using HPLC-grade water (A) and acetonitrile (B). The purified product in powder form was obtained after evaporation of the solvent and lyophilization. Nuclear magnetic resonance (NMR) analysis of the obtained product dissolved in pyridine-d5 (Sigma-Aldrich, USA) was done using a Bruker-BioSpin Avance 900 MHz NMR Spectrometer (Germany) for elucidation of the structure. One-dimensional NMR ([1]H-NMR, [13]C-NMR) and two-dimensional NMR (HSQC and HMBC) were performed to elucidate the structure of the product.

### 3.5. Biological Activity

Four cancer-cell lines (AGS, B16F10, HeLa, and U87MG) grown in RPMI medium, which contained fetal bovine serum (FBS, 10%), were used to test the anticancer activities of CK, Rh2, F12, and the glycoside product obtained from reaction of CK (CKGal), whose structures are shown in Figure 1. All cells were preserved at 37 °C in a humidified 5% $CO_2$ incubator. For the cell-growth assay, $2 \times 10^3$ cell/well in 96-well plates (SPL Life Sciences, Korea) were treated with the various concentrations (200 µmol, 100 µmol, 50 µmol, 25 µmol, 12.5 µmol, 6.25 µmol) of each compound for 72 h. Cell viability was measured by MTT colorimetric assay.

### 3.6. Statistical Analysis

The values are mean ± standard deviation (SD). SDs were calculated from the result of three independent experiments. A *p* value < 0.05 was considered to be statistically significant.

## 4. Conclusions

In conclusion, a regiospecific galactosyltransferase (LgtB) was able to convert CK into a new ginsenoside, 20-*O*-[*β*-D-lactopyranosyl]-20(S)-protopanaxadiol, which was a biologically active compound. A total of 392 mg/L (0.5 mmol) of CKGal was produced by this method within 3 h. The ginsenoside CK showed very good inhibitory activities against B16F10 and AGS cells compared to the other three compounds tested. This result suggests that CKGal can significantly lessen the cell viability of various cancer-cell lines, and can be a potential compound for further pharmaceutical studies in food, drugs, and cosmetics.

**Supplementary Materials:** The following are available online at http://www.mdpi.com/2076-3417/9/5/893/s1, Figure S1: NMR analysis of ginsenoside 20-O-[β-D-lactopyranosyl]-20(S)-protopanaxadiol (CKGal) in Pyridine-d5; Figure S2: NMR analysis of ginsenoside 20-O-[β-D-glucopyranosyl]-20(S)-protopanaxadiol (CK) in Pyridine-d5; Table S1: Cancer cell line cytotoxicity (μmol). Assessment of anticancer activity of compounds Rh2, 3-O-β-D-glucopyranosyl-20(S)-protopanaxadiol; F12, 3-O-β-D-glucopyranosyl-12-O-β-D-glucopyranosyl-20(S)-protopanaxadiol; CK, 20-O-beta-D-glucopyranosyl-20(S)-protopanaxadiol; and CKGal, 20-O-beta-D-lactopyranosyl-20(S)-protopanaxadiol against gastric carcinoma (AGS), skin melanoma (B16F10), cervical carcinoma (HeLa) and brain carcinoma (U87MG). (IC50; μmol, Mean ± SD, n = 3).

**Author Contributions:** S.D., R.P.P. and J.K.S. conceived and designed the experiments; S.D., B.S., S.L. and Y.J.K. performed the experiments; S.D., B.S. and R.P.P. analyzed the data, H.J.J, and J.K.S., contributed reagents/materials/analysis tools; S.D. wrote the paper.

**Funding:** This research was funded by the Next-Generation BioGreen 21 Program (SSAC, grant#: PJ013137) and Rural Development Administration, Republic of Korea.

**Acknowledgments:** The Next-Generation BioGreen 21 Program (SSAC, grant#: PJ013137) and Rural Development Administration, Republic of Korea supported this work.

**Conflicts of Interest:** Authors do not have any conflict of interest to declare.

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
