# Peer review of "Enzymatically Synthesized Ginsenoside Exhibits Antiproliferative Activity in Various Cancer Cell Lines"

_applsci, doi:10.3390/app9050893_

Round 1

Reviewer 1 Report

Major comments;

Cell type

Regarding to in vitro experiments of  each cancer, two kinds or more cell lines are appropriate. Authors should consider the key issue.

IC50 or pIC50

How did authors determine concentration of  compounds? Have authors ever detected IC50 or pIC50?

In vivo experiments

Whether authors could discuss in vivo animal experiments, it's important.

Language style

There are too many types and grammar mistakes in the current manuscript. Please double check before submission.

Author Response

1. Cell type

Regarding to in vitro experiments of each cancer, two kinds or more cell lines are appropriate. Authors should consider the key issue.

Response: Thank you for your comment. We will consider this issue in the future but right now we have limited cell lines available. Thus, in this study, we aimed to evaluate the anticancer effects of the compounds using various kinds of cancer cell lines.

2. IC50 or pIC50

How did authors determine concentration of compounds? Have authors ever detected IC50 or pIC50?

Response: The concentration of compounds (IC50) were determined based on the previous reports regarding to the anticancer effects of Rh2 and CK (Lee, S.; Kwon, M.C.; Jang, J.P.; Sohng, J.K.; Jung, H.J., The ginsenoside metabolite compound K inhibits growth, migration and stemness of glioblastoma cells. Int J Oncol. 2017, 51, 414-424). This article is cited in the text when necessary.

3. In vivo experiments

Whether authors could discuss in vivo animal experiments, it's important.

Response: Thank you for your valuable comment. In vivo animal experiment is important. We have not conducted the in vivo animal experiment with these compounds till now. In the future we have plan to study the therapeutic effects in animal models and study the underlying molecular mechanism of the compounds.

4. Language style

There are too many types and grammar mistakes in the current manuscript. Please double check before submission.

Response: The manuscript has been double checked for typos and grammatical errors have been rectified.

Reviewer 2 Report

The manuscript by Darsandhari et al. describes the enzymatic synthesis and the growth inhibitory activity of ginsenoside CKGal. The ginsenoside was chemically characterized by various analytical tools and biologically tested in four cell lines. The manuscript sufficiently describes the experiments, yet it does not adequately elaborate/articulate the impact of the work. Following comments should be addressed before publication:

1.    Page 3: Line 88, “described previously”, where?? Please provide a reference.

2.    Page 6: The authors state that “The improvement in such a biological property may have resulted from the presence of a different sugar moiety (lactose) instead of glucose in the aglycon.”

How adding one extra sugar unit would potentially enhance the biological activity? What is the potential molecular mechanism of CKGal? What is the rationale for such assumption?

Do you mean glycone or aglycone?

3.    What would be the potential molecular target (or pathway) for such molecules? Would it be intracellular, surface protein, or extracellular, in general? How adding one more sugar unit would impact the accessibility of CKGal to the target?

4.    The authors should add a paragraph describing potential structure-activity relationship considering the activity of CK, CKGal, Rh2, and F12.

5.    What is the potential impact of this work? The authors should provide a paragraph about the future directions built upon using the outcome of this work.

Author Response

The manuscript by Darsandhari et al. describes the enzymatic synthesis and the growth inhibitory activity of ginsenoside CKGal. The ginsenoside was chemically characterized by various analytical tools and biologically tested in four cell lines. The manuscript sufficiently describes the experiments, yet it does not adequately elaborate/articulate the impact of the work. Following comments should be addressed before publication:

Response: We are thankful to the reviewer for their evaluation. The reviewer suggested for the adequate elaboration of the work. We have tried our best to appropriately address the comments. We hope it to be convincing.

1.      Page 3: Line 88, “described previously”, where?? Please provide a reference.

Response: Appropriate change has been made in the text and highlighted in blue.

2.    Page 6: The authors state that “The improvement in such a biological property may have resulted from the presence of a different sugar moiety (lactose) instead of glucose in the aglycon.”

How adding one extra sugar unit would potentially enhance the biological activity? What is the potential molecular mechanism of CKGal? What is the rationale for such assumption?

Do you mean glycone or aglycone?

Response: It is just our assumption that the presence of different sugar moiety may have improved the biological activity of the compound. In the sentence “The improvement in such a biological property may have resulted from the presence of a different sugar moiety (lactose) instead of glucose in the aglycon.” The aglycone refers to protopanaxadiol. The change has been made in the manuscript in page 6, line 146.

While ginsenosides exhibits therapeutic activity, attachment of the sugar units enhance the solubility of compounds. Glycosides act as a key component for improving various pharmacological actions such as anticancer, antifungal, antitumor activities (Salas J.A.; Méndez, C., Engineering the glycosylation of natural products in actinomycetes. Trends Microbiol. 2007, 15, 219-32.)

The potential molecular mechanism of CKGal is yet to be explored. Components of ginseng have been found to induce cell cycle arrest in the G1 phase by upregulating the expression of p21.  Extracts from ginseng have been found to inhibit cancer cell growth either through suppression of the MAPKs pathway or through activation of MAPKs. Ethanol extract of ginseng was found to downregulate the expression of proliferating cell nuclear antigen (PCNA) in vivo and the activated form of Stat-3 in vitro. Compound K is known to upregulate the expression of p21 and activate c-jun N-terminal kinase (JNK) for cell cycle arrest in G1 phase in human monocytic leukemia cells.

Although we did not study the mechanism of action of such molecule, the potential mechanism of action of such compounds may be involved with the activities involved in the regulation of cell cycle, apoptosis or proliferation of cells.

Addition of sugar moieties increase the solubility of a compound. Hence, it may be possible that it is easier for the compound to reach the target because of the increase in its solubility.

Detail references is given in following question raised by the reviewer.

3. What would be the potential molecular target (or pathway) for such molecules? Would it be intracellular, surface protein, or extracellular, in general? How adding one more sugar unit would impact the accessibility of CKGal to the target?

Response: Components of ginseng have been found to induce cell cycle arrest in the G1 phase by upregulatin the expression of p21 (Nag, S.A. et.al, 2012).Extracts from ginseng were found to inhibit cancer cell growth through suppression of the MAPKs pathway (Ahn, J.Y. et.al 2006) or and also through activation of MAPKs (Wong, V.K.W. et al 2010). EAG was found to downregulate the expression of proliferating cell nuclear antigen (PCNA) in vivo and the activated form of Stat-3 in vitro (Wong, V.K.W. et al 2010). Compound K is known to upregulate the expression of p21 and activate c-jun N-terminal kinase (JNK) for cell cycle arrest in G1 phase in human monocytic leukemia cells (Kang, K.A. et al 2005).

Although we did not study the mechanism of action of such molecule in our anti-cancer study, the potential mechanism of action of such compounds may be involved with the activities involved in the regulation of cell cycle, apoptosis or proliferation of cells. Addition of sugar moieties increase the solubility of a compound. Hence, it may be possible that it is easier for the compound to reach the target because of the increase in its solubility.

4. The authors should add a paragraph describing potential structure-activity relationship considering the activity of CK, CKGal, Rh2, and F12.

Response: The anticancer activity of ginsenosides is better when the sugar unit is found in C-3 or C-20 when compared to sugar unit in C-6 (Nag, S.A. et.al, 2012). In our result with anticancer test, CK and Rh2 showed similar activity against all the cancer cells tested. The anticancer activity of both of these ginsenosides were better than that of F12. CKGal improved the activity of CK in case of B16F10 and AGS cancer cell lines. However, the activity was not improved HeLa and U87mG cancer cell lines that we tested. The reason is unclear and warrants further study.  

5. What is the potential impact of this work? The authors should provide a paragraph about the future directions built upon using the outcome of this work.

Response: The number of sugar moieties and different substituent groups in the side chain of the dammarane saponin or aglycone protopanaxadiol affect the activity of the ginsenosides against different cancer cell lines. Our result will be helpful for evaluating the SAR of other ginsenosides. Our finding may be useful for the development of novel agents for treatment of various cancers.

Round 2

Reviewer 1 Report

Authors answered question in a satisfactory manner, generally the quality of manuscript has been improved.

Reviewer 2 Report

The authors adequately responded to my comments. No further suggestions.